

# Reliability of dynamic and isometric upper muscle strength testing in breast cancer survivors

Wanderson Santos[1], Vitor Marques[1], Claudio Andre B. de Lira[1], Wagner Martins[2], Amilton Vieira[2], Diba Mani[3], Claudio Battaglini[4] and Carlos Vieira[1]

[1] College of Physical Education and Dance, Federal University of Goias-UFG, Universidade Federal de Goias, Goiania, Goias, Brazil
[2] College of Physical Education, University of Brasilia-UnB, Professor, Brasilia, Distrito Federal, Brazil
[3] Department of Applied Physiology and Kinesiology, University of Florida, Gainesville, Florida, United States
[4] Department of Exercise and Sport Science and Lineberger Comprehensive Cancer Center, University of North Carolina at Chapel Hill, Chapel Hill, North Carolina, United States

Corresponding author
Carlos Vieira, vieiraca11@gmail.com

## ABSTRACT

Breast cancer is the most common cancer in women worldwide, and its treatment usually involves a combination of many medical procedures, including surgery, chemotherapy, radiotherapy, and hormonal therapy. One of the detrimental effects on physical function is reduced upper limb muscle strength. This study aimed to evaluate upper body strength intra-day and inter-day (test-retest) reliability using the handgrip strength test (HGS) and the bilateral isometric bench press (BIBP) and the test-retest reliability of the one repetition maximum on the bench press (BP-1RM) in breast cancer survivors (BCS). Thirty-two (52.94 ± 8.99 yrs) BCS participated in this study. The muscle strength tests were performed in two different moments, three to seven days apart. Intraclass coefficient correlation (ICC) and coefficient of variation (CV) were used to assess the reliability. Standard error of measurement (SEM), typical error of measurement (TEM), and minimally detectable change (MDC) analyses were performed. The Bland-Altman analysis was used to assess the agreement between test-retest. We found a reliability that can be described as "high" to "very high" (ICC ≥ 0.88; CV ≤ 10%) for intra-day and test-retest. SEM% and MDC% were lower than 5% and 11%, respectively, for all intra-day testing. SEM% and TEM% ranged from 3% to 11%, and MDC% ranged from 9% to 23% in the test-retest reliability. The agreement demonstrated a systematic bias ranging from 2.3% to 6.0% for all testing, and a lower systematic bias may be presented in the non-treated side assessed by HGS and BIBP. HGS, BIBP, and BP-1RM assessments are reliable for measuring upper-body muscle strength in BCS.

## INTRODUCTION

Breast cancer is the most common cancer in women worldwide, estimated at 2.26 million new cases and 685 thousand deaths in 2020 (*Ferlay et al., 2021*). Breast cancer treatment usually involves a combination of many medical procedures, including surgery, chemotherapy, radiotherapy, and hormonal therapy. Combined or not, these treatments could negatively influence psychological, physical, and overall functionality. One of the detrimental effects on physical function is reduced upper limb muscle strength, especially for those muscles crossing the shoulder joint (*Harrington et al., 2011*). Considering that greater muscle strength is an essential functional capacity and one predictor for improved quality of life and increased functional capacity in cancer survivors (*Hanson et al., 2016*; *dos Santos et al., 2017*; *Harrington et al., 2013*), these functions must be assessed and recovered as soon as possible. Addressing the issue of reduced upper limb strength often observed throughout treatment is paramount.

Dynamic or isometric muscle contractions may be performed to assess the maximum muscle strength in breast cancer survivors (BCS) (*dos Santos et al., 2017*; *Heyward & Gibson, 2014*). The American College of Sports Medicine (ACSM) recommends the bench press to assess upper limb muscle strength dynamically (*American College of Sports Medicine, 2018*). The one repetition maximum on the bench press (BP 1-RM) has been used to measure muscle strength in breast cancer studies (*Rogers et al., 2017*; *Winters-Stone et al., 2012*). Although the safety of this measure has been reported, there is a lack of information regarding its reliability.

Arguments advocating the advantage of isometric muscle contraction testing over dynamic contractions include reduced time to learn the task, easier to guide and set the individual's posture, and a minimal technical requirement to perform the movement (*Drake, Kennedy & Wallace, 2017*). Specifically, the isometric bench press assessment has been reported to be highly correlated with the dynamic bench press assessment, as evaluated by a 1-repetition maximum (*Murphy et al., 1995*). Therefore, the expansion and enhancement of increasing the scientific evidence exploring the use of the isometric bench press test in oncological populations, such as BCS, is imperative and essential as it could provide diagnostics regarding maximum force production as well as upper limb muscle weakness and muscle strength asymmetry.

To our knowledge, few studies have explored using isometric muscle contraction strength tests to measure and assess upper limb strength in BCS. *Hagstrom, Shorter & Marshall (2019)* implemented a compound movement, a unilateral-inclined bench press. When comparing both upper limbs in BCS, they found that the muscle peak force output in the ipsilateral upper limb (treated side) was lower than the contralateral upper limb (non-treated side). On the other hand, *Bertoli et al. (2020)* found no difference in muscle peak force output on the treated side compared to the non-treated side using an isolated movement, shoulder strength movement abduction at 65°. These BCS studies reported high intra-test reliability (*Hagstrom, Shorter & Marshall, 2019*; *Bertoli et al., 2020*), although no information on test-retest reliability and agreement has been reported for BCS.

Regarding the testing reliability in BCS, reporting the reliability of just 1 day of testing (intra-day) could provide different reliability results in comparison to multiple days (test-retest) (*Sorbie et al., 2018*; *van Schooten et al., 2013*; *Merlet et al., 2018*). In addition, muscle weakness, fear of movement, impairment in the shoulder joint range of motion, lymphedema, and fatigue have been previously cited and reported as long-term side effects of breast cancer treatment (*Chan, Lui & So, 2010*; *Hayes et al., 2010*). Those side effects have been shown to impair upper body strength (*dos Santos et al., 2017*) in BCS. Therefore, it is crucial to evaluate dynamic and isometric maximum muscle strength reliability through intra-day and inter-day testing to assess upper body strength in BCS accurately.

This study aimed to evaluate upper body strength intra-day and inter-day (test-retest) reliability using the handgrip strength test and the bilateral isometric bench press (BIBP) and the test-retest reliability of the 1-RM on the bench press in BCS. The secondary aim was to report, as Supplemental Material and subset analysis, the influence of lymphedema, mastectomy, and limb dominance on the reliability of all tests.

## MATERIALS AND METHODS

### Design and participants

The study comprised two separate visits to our laboratory, spaced 3 to 7 days apart. On the first testing day, the patients completed a survey, underwent anthropometric measurements, and participated in a familiarization session. After 5 min of passive rest, the patients performed the upper body strength testing in the following order: handgrip, bilateral isometric bench press, and bench press one-repetition maximum (1-RM). During the second testing day (retest), all three tests were performed in the same order. The patients performed the less effort-demanding isometric tests before the dynamic 1-RM bench press to minimize fatigue (*Heyward & Gibson, 2014*; *Haff & Dumke, 2019*). In both testing days, the rest interval between tests was 5 min.

The study included 32 BCS women, assessed between November 2019 and October 2021, aged between 35 and 67 (52.94 ± 8.99 years old), who participated in this reliability and agreement study. Due to the number of participants, we chose not to separate into groups by age group. The eligibility criteria included that the subjects were confirmed within breast cancer diagnosis stages I to III, underwent breast cancer therapy including breast cancer surgery, and were not involved in any regular exercise program in the last 6 months. The participants who underwent breast reconstruction were not excluded from this study. BCS patients were excluded from the study if they had musculoskeletal limitations, such as upper limb disability, that could compromise exercise performance and/or any uncontrolled chronic disease, such as cardiovascular diseases (*i.e.*, hypertension, they have had heart attack or stroke), diabetes or neuromotor diseases that could represent risk for an adverse event during the study.

The participants were recruited through personal contact (phone call or face-to-face), medical referral, and an advertising poster on the Clinical Hospital of the Federal University of Goias (Goiania, Goias, Brazil). Table 1 presents the sociodemographic, cancer treatment status, and anthropometric characteristics of the participants.

| Table 1 Characteristics. | |
|---|---|
| **Characteristics** | **N = 32** |
| Age (year)–mean (SD) | 52.94 (8.99) |
| Education–no. (%) | |
| <8 years of the study | 16 (50) |
| >8 years of the study | 16 (50) |
| Self-reported race–no. (%) | |
| Caucasian | 12 (37.5) |
| Non Caucasian | 20 (62.5) |
| Occupation–no. (%) | |
| Homemaker or cleaner | 5 (15.63) |
| Retired | 7 (21.88) |
| Hairdresser or manicure | 2 (6.25) |
| Dressmaker | 3 (9.38) |
| Housewife | 6 (18.75) |
| Administrative or education assistant | 3 (9.38) |
| Health professionals | 3 (9.38) |
| Street market worker or businesswoman | 3 (9.38) |
| Marital status–no. (%) | |
| Single | 9 (28.12) |
| Married | 16 (50) |
| Divorced | 3 (9.38) |
| Widow | 4 (12.5) |
| Arterial hypertension–no. (%) | 8 (25) |
| Months since cancer diagnosis–mean (SD) | 54.91 (44.32) |
| Cancer stage–no. (%) | |
| I | 8 (25) |
| II | 10 (31.25) |
| III | 14 (43.75) |
| Breast surgery–no. (%) | |
| Mastectomy | 15 (46.87) |
| Quadrantectomy | 17 (53.13) |
| Breast reconstruction | 5 (15.62) |
| Breast surgery side–no. (%) | |
| Right | 19 (59.37) |
| Left | 13 (40.63) |
| Months since breast surgery–mean (SD) | 44.88 (38.27) |
| Chemotherapy–no. (%) | 31 (96.88) |
| Adjuvant | 21 (67.74) |
| Neoadjuvant | 10 (32.26) |
| Radiotherapy–no. (%) | 30 (93.75) |
| Hormone therapy–no. (%) | 29 (90.62) |
| Tamoxifen | 20 (68.97) |

| Characteristics | N = 32 |
|---|---|
| Aromatase inhibitors | 9 (31.03) |
| Self-reported lymphedema–no. (%) | 12 (37.5) |
| Anthropometry | |
| Weight (kg)–mean (SD) | 70.04 (14.60) |
| Height (m)–mean (SD) | 1.57 (0.06) |
| BMI–mean (SD) | 28.20 (5.32) |
| Level physical activity* (MET-h/wk)–mean (SD) | 9.64 (11.53) |

**Note:**
* Assessed by International Physical Activity Questionnaire (IPAQ); SD, standard deviation; BMI, body mass index; MET, metabolic equivalent of task.

The study was approved by the Research Ethics Committee of the Federal University of Goiás (CAAE: 50717115.4.0000.5083) and by the Research Ethics Committee of the Clinical Hospital of the Federal University of Goias (CAAE: 50717115.4.3001.5078). All BCS volunteers provided written consent before participating in the study.

## Procedures

### Handgrip strength test

The HGS test followed the guidelines of the American Society of Hand Therapists (*Fess, 1983*). The tests and re-tests consisted of three attempts with 3–5 s of maximal voluntary contraction with 90° elbow, with verbal stimulus, alternating right and left hands with 1 min of rest interval between attempts (*Fess, 1983*). The BCS patients were seated with their feet on the ground and their backs resting on a chair, with a 90° angle to the hip and knee joints. A digital dynamometer (model EH101, E.clear®, Berlin, Germany) was used. The three best attempts of the five attempts of each side were used to calculate the reliability intra and inter-days (test-retest) reliability.

### Bilateral isometric bench press test

The BIBP test was performed on a custom-made hydraulic bench press (Figs. 1A–1C). Two load cells were connected on a customized loading pin for the free weight plates, with handles attached on each side (Figs. 1A–1D). The bench included a vertical adjustment with the load cells placed below the elbows (Figs. 1C). Using a manual goniometer, the elbow flexion was set at ~90° (Figs. 1B and 1C). Based on the strongest position, the initial bench press position was self-selected (Fig. 1D). The warm-up consisted of three trials performed at 50%, 75%, and 100% perceived effort, lasting three seconds per trial. A 1-min rest period was provided between trials. After completing the warm-ups, the BCS participants rested for an additional 3 min after the warm-up and then started the BIBP test. Before each trial on the BIBP, the BCS participants were instructed and directed to "lift harder and as fast as possible, in three, two, one, and go!" command. Strong verbal encouragement was given during each trial. The participants performed three to four of these maximal voluntary isometric contractions, lasting 5 s each. If either limb had a

Bilateral Isometric Bench Press Test

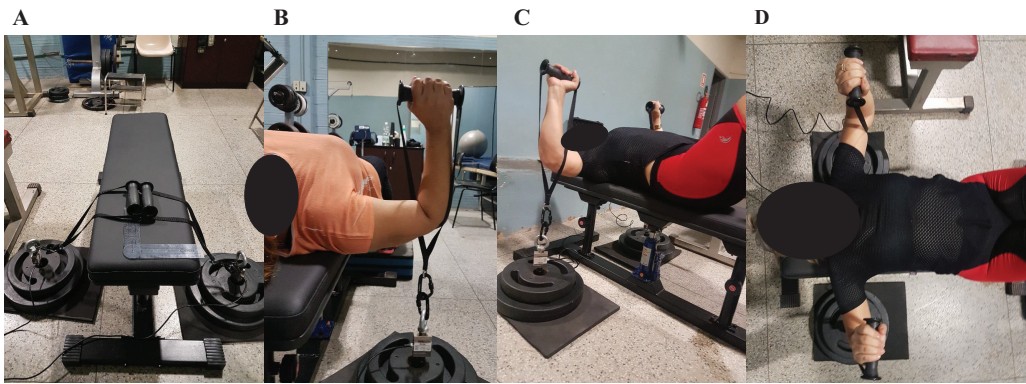

**Figure 1** **Bilateral isometric bench press.** (A) Customized bench; (B) elbow flexed ~90°; (C and D) "strongest position"; (B–D) load cells placed below elbows.

calculated coefficient of variation (CV) greater than 10% between the three first trials, a fourth trial was recorded, and the lowest trial was deleted and discarded. A 3-min rest interval was given between trials. The average of the three best trials was used to assess the peak force output reliability for the treated and non-treated sides and their sum (treated side plus non-treated side). The warm-ups and maximal voluntary isometric contractions were repeated at the retest visit.

### Data acquisition

Two load cells (S-Type Model OP-312; OPTIMA SCALE, Rancho Cucamonga, CA, USA) with a 750-lb max force capacity were used in the study. A custom software designed in LabVIEW (National Instruments Corp., Austin, TX, USA) performed all digital signal processing. This software received two channel signals from designated electronic hardware based on the ESP32 microcontroller. The signal, sampled at 80 Hz, was transmitted *via* WiFi and then analyzed in real-time on a laptop. Before each day of testing, the system was calibrated using known masses. Force was expressed in Newtons (N) and converted to kilogram-force (Kgf) to report the total normalized muscle peak force output.

### The one-repetition maximum test

The 1-RM test and retests were performed using the bench press exercise with free-weight, plate-loaded (Supplemental Material). The exercise technique followed the recommended standardized procedures set by the National Strength and Conditioning Association (NSCA) (*National Strength and Conditioning Association, 2016*). During the 1-RM test and retest, two experienced exercise science professionals supervised the BCS participants. The same exercise science professionals managed all the measurements throughout the study. The BCS participants underwent three to five 1-RM attempts for each exercise trial.

The warm-up performed included one set of ten repetitions with an empty bar, the barbell weighted 11 kg; one set of three repetitions with ~60% of 1-RM of perceived load; one set of three repetitions with ~80% of 1-RM perceived load; and one set with two repetitions with ~90% of 1-RM perceived load. The rest interval during the warm-up was 2

min. After the last load used during the warm-up, BCS rested 3 min before beginning the 1-RM test. Three to five attempts were allowed to reach the maximum weight lifted with one repetition. The movement velocity was orientated to perform with maximum voluntary velocity during the concentric action. The rest interval between attempts was 5 min. The retest day was performed 3 to 7 days after the test day, using the load reached on the test day as a reference (*Heyward & Gibson, 2014*; *Haff & Dumke, 2019*).

## Statistical analyses

The sample size was calculated using a web-based sample size calculator for reliability studies (*Arifin, 2018*). The calculation was set using $\alpha$ = 0.05, power (1-$\beta$) = 80%, k = 2 and 3 (two measures in the test-retest (one measure for each day) and three trials in the intra-day measures), and a minimum acceptable reliability 0.8. The expected dropout rate was 0 participants. The final sample size was 5 and 8 for intra-day and test-retest reliability, respectively.

Descriptive statistics were presented as mean and standard deviation (SD). Intra-day and test-retest were used the intraclass coefficient correlation (ICC) and coefficient of variation (CV), which was calculated as: $CV = \left(\dfrac{SD}{mean}\right) \times 100$, for reliability were used to assess intra-day and test-retest reliability. CV represents the within-subject variability, both intra-day and inter-day. Intra-day CV was calculated in two moments over three trials, one during the test day and another during the retest day. Inter-day CV was derived from the variability between days as two measures: test-retest. The ICC method implemented was a two-way mixed effect, with an average of two (test-retest) or three (intra-day) measurements and absolute agreement (*Koo & Li, 2016*). The ICC and CV are presented as mean and 95% confidence intervals (CI).

The analyses of measurement absolute and relative error of measurement were investigated using the standard error of measurement ((SEM); SEM absolute = SD of the average of trials (intra-day) or test-retest (inter-day) scores multiplied by the square root of 1 – ICC; (SEM%) SEM relative = SEM absolute score divided by the average of trials (intra-day) or test-retest (inter-day) scores and multiplying by 100), typical error of the measurement ((TEM); TEM absolute, SD of the mean difference divided by the square root of 2; ((TEM%) TEM relative, TEM absolute score divided by the average of the test-retest score and multiplying by 100), and minimally detectable change ((MDC); MDC absolute = 1.96 × the square root of 2 × SEM; (MDC%) MDC relative = MDC absolute score divided by the average of trials (intra-day) or test-retest (inter-day) scores and multiplying by 100) (*Weir, 2005*).

*Bland & Altman (1999)* plots were created to evaluate the agreement between the test and retest trials. Munro's reliability classification was used to interpret the ICC coefficients: a coefficient between 0.50 to 0.69 reflects moderate correlation, 0.70 to 0.89 reflects high correlation, and 0.90 to 1.00 indicates very high correlation.

Statistical analyses were performed using R (version 4.0.3; *R Core Team, 2020*) through the RStudio interface (version 1.3.1093; *RStudio Team, 2021*) and Microsoft Excel (2010).

**Table 2 Analysis of intra-day reliability and error of measurement on the handgrip strength and bilateral isometric bench press in breast cancer survivors.**

| | Trial 1 (mean ± SD) | Trial 2 (mean ± SD) | Trial 3 (mean ± SD) | ICC (95% CI) | CV (95% CI) | SEM (SEM%) | MDC (MDC%) |
|---|---|---|---|---|---|---|---|
| *HGS test (kgf)* | | | | | | | |
| Treated side | 24.94 (6.41) | 24.88 (6.86) | 25.08 (6.31) | 0.98 [0.98–0.99] | 4.90 [3.33–6.46] | 0.68 (2.71) | 1.87 (7.51) |
| Non-treated side | 24.67 (5.34) | 24.97 (5.39) | 24.74 (5.03) | 0.99 [0.98–0.99] | 3.76 [2.96–4.56] | 0.57 (2.29) | 1.58 (6.36) |
| *HGS retest (Kgf)* | | | | | | | |
| Treated side | 25.87 (5.27) | 26.20 (5.20) | 26.17 (5.37) | 0.99 [0.97–0.99] | 3.27 [2.14–4.39] | 0.59 (2.28) | 1.64 (6.32) |
| Non-treated side | 25.54 (5.18) | 25.53 (4.94) | 25.80 (5.10) | 0.98 [0.97–0.99] | 3.71 [2.87–4.53] | 0.59 (2.31) | 1.64 (6.41) |
| *BIBP test (N)* | | | | | | | |
| Treated side | 99.65 (22.23) | 99.56 (21.15) | 97.45 (21.33) | 0.98 [0.96–0.99] | 4.36 [3.23–5.49] | 2.85 (2.87) | 7.89 (7.98) |
| Non-treated side | 109.92 (28.32) | 107.31 (26.34) | 105.14 (25.55) | 0.97 [0.95–0.99] | 5.33 [3.83–6.83] | 4.14 (3.84) | 11.47 (10.67) |
| Total | 209.57 (48.99) | 206.87 (45.90) | 202.59 (45.47) | 0.98 [0.97–0.99] | 4.19 [2.90–5.48] | 5.88 (2.98) | 16.31 (8.26) |
| *BIBP retest (N)* | | | | | | | |
| Treated side | 103.98 (25.42) | 107.01 (27.50) | 106.69 (26.61) | 0.98 [0.96–0.99] | 4.78 [3.67–5.89] | 3.48 (3.28) | 9.65 (9.11) |
| Non-treated side | 107.75 (24.95) | 108.76 (26.25) | 111.12 (25.79) | 0.97 [0.95–0.99] | 5.17 [3.97–6.37] | 4.11 (3.76) | 11.40 (10.43) |
| Total | 211.74 (47.59) | 215.76 (51.40) | 217.82 (50.07) | 0.98 [0.96–0.99] | 4.21 [3.19–5.21] | 6.27 (3.04) | 17.38 (8.43) |

Note:
HGS, handgrip strength; BIBP, bilateral isometric bench press; *N*, Newton; SD, standard deviation; CV, coefficient of variation; ICC, intra-class coefficient correlation; CI, 95% confidence intervals; SEM, standard error of measurement; MDC, minimally detectable change.

## RESULTS

### Intra-day reliability

The intra-day testing for both the handgrip strength and BIBP showed very high reliability of the upper limbs and acceptable error of measurement, with the ICCs ranging from 0.97 to 0.99. ICC's *posteriori* power test (1-β) resulted in 0.99 for all variables. The CV ranged from 3% to 5%. The SEM % was lower than 4%, and the MDC % was ≤11%. The intra-day reliability and the analysis of error are presented in Table 2 and Supplemental Materials 1, 3, and 5.

### Test-retest reliability and agreement

All muscle strength test assessments (test-retest) presented high to very high reliability for the upper limbs. The ICC ranged from 0.85 to 0.96. ICCs *posteriori* power test (1-β) resulted in 0.99 for all variables. The test-retest CV ranged from 3% to 10%. The reliability results are presented in Table 3, Supplemental Materials 2, 4, and 6. The analyses of error, relative and absolute SEM, TEM, and MDC are presented in Table 3.

The Bland-Altman plots reveal the systematic bias with a 95% CI of limits of agreement (Fig. 2). The relative mean difference between the test and the retest presented were as follows: HGS treated side, 5.53% CI [1.09–9.97]; HGS non-treated side, 2.34 % CI [−0.33 to 5.01]; BIBP treated side, 5.99% CI [1.01–10.98]; BIBP non-treated side, 2.68% CI [−2.68 to 8.03]; BIBP summed values, 4.34% CI [−0.37 to 9.05]; and, 1-RM, 5.17% CI [2.19–8.16]. The test-retest agreement regarding the influence of lymphedema, breast cancer surgery (mastectomy), and limb dominance are shown in Figs. S1–S6.

**Table 3 Analysis of test-retest reliability and error of measurement of upper body strength testing in breast cancer survivors.**

| | Test (mean ± SD) | Retest (mean ± SD) | ICC (95% CI) | CV % (95% CI) | TEM (TEM%) | SEM (SEM%) | MDC (MDC%) |
|---|---|---|---|---|---|---|---|
| *HGS (Kgf)* | | | | | | | |
| Treated side | 24.69 (5.84) | 25.76 (4.90) | 0.94 [0.87–0.97] | 6.02 [3.36–8.68] | 1.63 (6.48) | 1.27 (5.04) | 3.52 (13.98) |
| Non-treated side | 25.17 (4.93) | 25.67 (4.69) | 0.97 [0.94–0.98] | 4.27 [3.06–5.49] | 1.16 (4.57) | 0.85 (3.37) | 2.37 (9.33) |
| *BIBP (N)* | | | | | | | |
| Treated side | 95.21 (20.86) | 101.63 (24.73) | 0.88 [0.73–0.95] | 8.66 [6.48–10.84] | 9.75 (9.91) | 7.81 (7.93) | 21.64 (21.99) |
| Non-treated side | 102.17 (25.62) | 104.53 (23.92) | 0.88 [0.76–0.94] | 7.75 [5.14–10.35] | 11.41 (11.04) | 8.42 (8.14) | 23.34 (22.58) |
| Total | 197.38 (45.12) | 206.17 (46.72) | 0.90 [0.79–0.95] | 7.65 [5.05–10.25] | 19.13 (9.48) | 14.55 (7.21) | 40.33 (19.99) |
| *1-RM (Kg)* | 22.75 (5.11) | 23.81 (4.58) | 0.95 [0.86–0.98] | 4.10 [2.10–6.10] | 1.24 (5.34) | 1.02 (4.41) | 2.84 (12.23) |

**Note:**

N, Newton; BIBP, bilateral isometric bench press; SD, standard deviation; CV, coefficient of variation; ICC, intra-class coefficient correlation; CI, 95% confidence interval; SEM, standard error of measurement; MDC, minimally detectable change.

## DISCUSSION

This study aimed to analyze and evaluate the intra-day and test-retest reliability and agreement for the isometric and dynamic maximum muscle strength testing in BCS. In evaluating the intra-day testing results, very high reliability (ICC ≥ 0.97) was verified by a fairly low CV (CV < 6%), SEM (absolute and relative), and MDC (absolute and relative). The test-retest evaluation also revealed high to very high reliability and agreement with an acceptable error of the measurement, absolute and relative TEM, SEM, and MDC. The intra-day analysis presented slightly higher ICC values and lower CV, SEM, and MDC than the test-retest analysis. Based on these findings and interpretation, maximum muscle strength testing using unilateral (HGS) or bilateral (BIBP) to assess maximum voluntary isometric contraction is likely a good and reliable muscle strength measurement in BCS-treated and non-treated upper limbs, as well as dynamic compound movement such as the bench press 1-RM.

To our knowledge, there is a lack of information regarding isometric muscle strength test-retest reliability in the BCS population. Intra-day reliability reported in a unilateral isometric inclined bench press test revealed an ICC of 0.89 (95% IC [0.8–0.95]) at baseline and 0.93 (95% IC [0.86–0.97]) after 17 weeks of resistance training (*Hayes et al., 2010*). *Bertoli et al. (2020)* reported a 5% CV of peak force in shoulder abduction at 65° in BCS, similar to our current study findings with high reliability with a 6% CV. However, intra-day reliability could not be used as a stable measure throughout the days. Our evaluation of the test-retest presented lower ICC, higher CV, SEM, and MDC than intra-days analysis. Similar findings have been found when comparing intra-days and test-retest reliability (*Sorbie et al., 2018*; *van Schooten et al., 2013*; *Merlet et al., 2018*). This could be explained by the learning effect of a task across trials, improving with each repeated performance. Therefore, we recommend incorporating at least 2 days of testing to minimize the error of measurement in studies evaluating muscle peak force output in BCS. However, these measurements present high intra-days reliability and are even better than test-retest, warranting further investigation into using these measurements to assess upper limb strength in BCS.

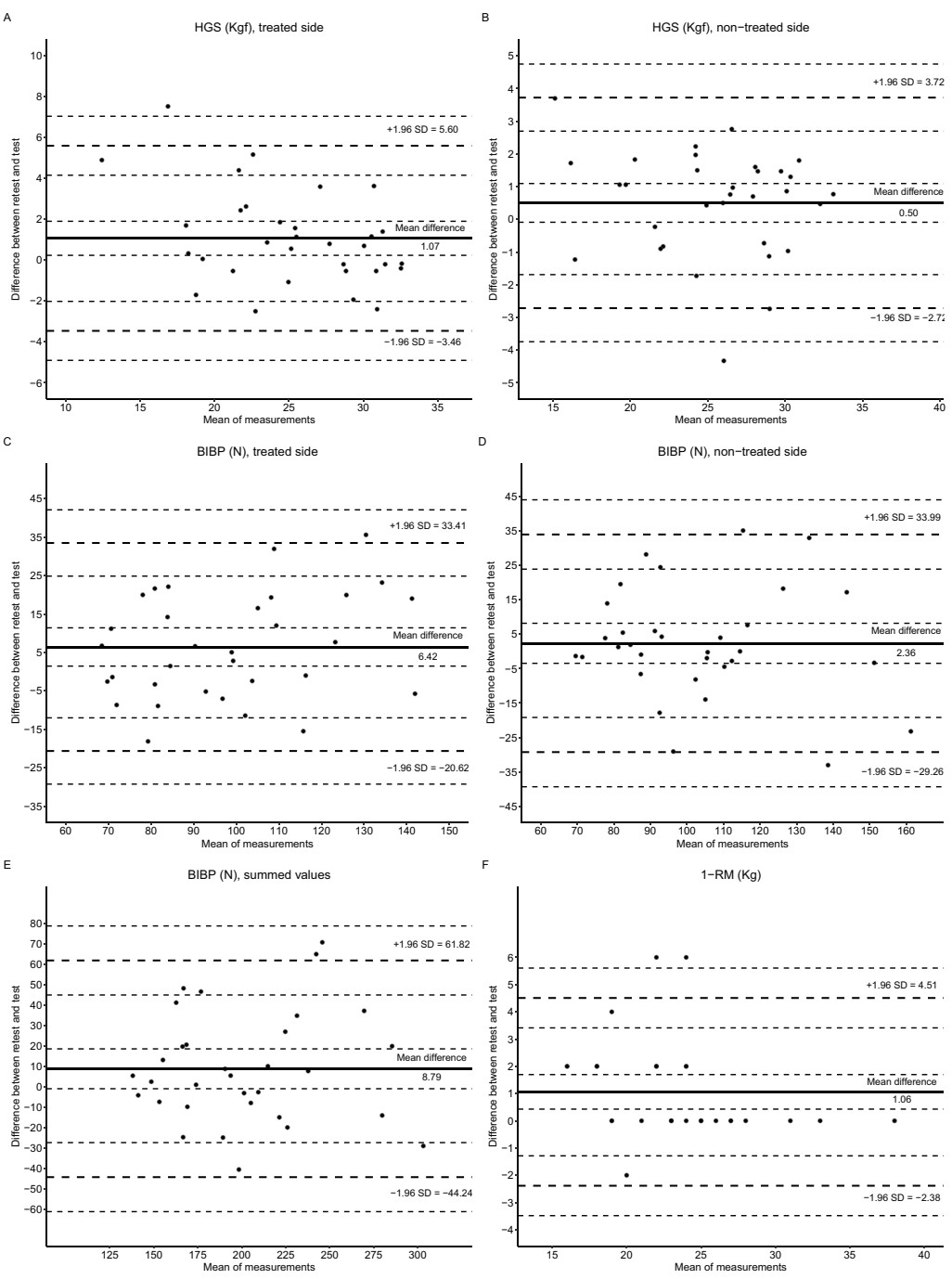

**Figure 2 Bland-Altman plots of maximum muscular strength tests: HGS, handgrip strength; BIBP, bilateral isometric bench press; and, 1-RM, one repetition maximum.** The dotted line (+1.96 SD and −1.96 SD) represents the limits of the agreement's upper and lower boundaries. The continuous line (mean difference) on the center of the plot represents the systematic bias. The continuous line on the Y axis represents the mean difference between retest and test, and on the X axis represents the mean of retest and test.

Traditionally, the isometric bench press test implemented using force plates on the Smith machine with elbow flexion at 90° in healthy populations has shown high reliability, demonstrated by an ICC of 0.89 (*Young et al., 2014*) and 0.93 (*Loturco et al., 2016*). Using a single load cell to measure the isometric bench press also showed high test-retest reliability, with an ICC of 0.94 (*Pinto et al., 2013*). Similarly, our study found a high to very high test-retest reliability when incorporating two load cells with acceptable CV ($\leq$10%). The advantage of using two load cells to perform a BIBP test is that the assessment of muscle strength bilaterally with data from each arm simultaneously could also be utilized as biofeedback as an important consideration for enhanced athletic performance and rehabilitation.

The bench press 1-RM has been used to assess dynamic upper body strength in breast cancer studies. However, to our knowledge, just one other project study has reported a CV of 7.5% on the bench press 1-RM test as the only criteria of reliability (*Winters-Stone et al., 2012*). That is similar to our current study, where CV (4.1%) was on the bench press 1-RM. Although the results on the retest day were superior to the test day, the intra-day measures presented a good agreement with a systematic bias of 1.06 kg (5.17%). Therefore, the bench press 1-RM test appears reliable for assessing upper body strength in BCS intra-day and test-retest.

Regarding secondary analysis performed to evaluate the influence of BCS treatment specific conditions for factors that may interfere in the reliability, such as lymphedema, type of breast cancer surgery (*i.e.*, mastectomy), and the influence when breast cancer surgery occurred on the dominant side or not, overall, the results. Even though these factors may affect the reliability, high test-retest reliability (ICC > 0.85, CV $\leq$ 10%) was achievable. However, BCS with lymphedema presented a higher variability in HGS and 1-RM. In addition, higher variability was shown when the breast cancer surgery happened in the non-dominant side assessed by HGS. Therefore, researchers and health professionals who are involved use in upper muscle strength testing must take into consideration to be aware of the influences on the reliability in BCS with that lymphedema and limb dominance appear to have on test reliability in BCS for a more precise evaluation of upper limb strength.

## STRENGTHS AND LIMITATIONS

The major strength of this study was the tight control and supervision of all tests by experienced sports science professionals. Another strength of this study was the 100% participation of all recruited BCS with no attrition issues. All BCS enrolled in the study participated in all testing trials, and no adverse event occurred throughout the study. This is an important finding since the study testing protocols appear safe and tolerable to all BCS who participated in the study. All days were implemented under the guidance of at least two sports science professionals, ensuring the BIBP test's accuracy. Some limitations of this study include the small sample size for the secondary analysis; future studies should consider a larger sample size with adequate power to explore the number of subjects recruited and also to investigate the influence of the type of surgery and breast cancer-related with lymphedema have in the reliability on of the others isometric and

dynamic muscle strength testing. Also, the time after completion of anti-cancer treatments was not controlled in this study and could have influenced the results of the current study to a certain extent.

## CONCLUSION

The isometric and dynamic maximum muscle strength testing used in this study (HGS, BIBP, and the 1-RM bench press test) showed to be a useful test with presented a high to very high rate of reliability and agreement, with an acceptable error of measurement in early-stage BCS who underwent breast cancer therapy including surgery and were not involved in any regular exercise program in the last 6 months. BCS with lymphedema may present a higher variability in HGS and 1-RM. Handgrip strength may present higher variability when breast cancer surgery happens on the non-dominant side. This study also showed that the BIBP test may be incorporated to assess the upper body muscle strength in a bilateral task in a clinical setting, enabling healthcare professionals to explore different ways to improve force production on the treated side or weaker arm in BCS.

### Funding
The authors received no funding for this work.

### Competing Interests
The authors declare that they have no competing interests.

### Author Contributions
- Wanderson Santos conceived and designed the experiments, performed the experiments, analyzed the data, prepared figures and/or tables, authored or reviewed drafts of the article, and approved the final draft.
- Vitor Marques conceived and designed the experiments, performed the experiments, analyzed the data, authored or reviewed drafts of the article, and approved the final draft.
- Claudio Andre B. de Lira conceived and designed the experiments, authored or reviewed drafts of the article, and approved the final draft.
- Wagner Martins conceived and designed the experiments, analyzed the data, authored or reviewed drafts of the article, and approved the final draft.
- Amilton Vieira conceived and designed the experiments, analyzed the data, authored or reviewed drafts of the article, and approved the final draft.
- Diba Mani conceived and designed the experiments, authored or reviewed drafts of the article, and approved the final draft.
- Claudio Battaglini conceived and designed the experiments, authored or reviewed drafts of the article, and approved the final draft.
- Carlos Vieira conceived and designed the experiments, performed the experiments, analyzed the data, prepared figures and/or tables, authored or reviewed drafts of the article, and approved the final draft.
## Ethics

The following information was supplied relating to ethical approvals (*i.e.*, approving body and any reference numbers):

The study was approved by the Research Ethics Committee of the Federal University of Goias (CAAE: 50717115.4.0000.5083) and by the Research Ethics Committee of the Clinical Hospital of the Federal University of Goias (CAAE: 50717115.4.3001.5078).

## Data Availability

The raw measurements are available in the Supplemental File.

## Supplemental Information

Supplemental information for this article can be found online at http://dx.doi.org/10.7717/peerj.17576#supplemental-information.

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
