# Peer review of "Reliability of dynamic and isometric upper muscle strength testing in breast cancer survivors"

_PeerJ, doi:10.7717/peerj.17576_

## Round 0.1 · original submission · Minor Revisions

After careful consideration, we feel that it has merit but does not fully meet PeerJ’s publication criteria as it currently stands. Therefore, we invite you to submit a revised version of the manuscript that addresses the points raised during the review process.

I invite the authors to revise the paper, taking into careful consideration the comments and recommendations provided by the reviewers.

Reviewer 1 ·

Basic reporting

Globally the English used throughout is clear and the authors reported adequate literature references.
The structure of the article is well organized as well as the figures and tables. Raw data are shared.
The manuscript contains relevant results to hypotheses, despite, as reported in the limitation, the sample size is small and heterogeneous.

Experimental design

Authors described methods with sufficient detail and information to replicate and the research question is well defined.

Line 117-121: Please explain the sequence of the test and the rest given to the patients between the different trial (the handgrip strength test, the bilateral isometric bench press test, and the one-repetition maximum on the bench press exercise). Comment on whether according to the authors performing the strength tests on the same day may have affected the performance of one of the tests.

Line 126: Please specify in the inclusion criteria that were included patients with different type of surgery.

Validity of the findings

All underlying data have been provided they are consistent despite the limitations.
Conclusions are well stated, linked to original research question & limited to supporting results.

Reviewer 2 ·

Basic reporting

Thank you for this valuable work. There are a few parameters that I think would strengthen the study if organized on behalf of basic reporting.

Information on where participants were referred from
The method by which the physical activity level specified in the tables was determined

Experimental design

No comment

Validity of the findings

The contribution of the study to the literature and the clinic has been mentioned, but its extension would be more enlightening for the readers.
Some of the descriptive data such as marital status and education level of the participants were not used in the analysis. I think that there may be variables that may not give an idea about the clinical status of the participants.

---

## Round 0.2 · accepted · Accept

Authors have addressed all the reviewers' comments and now the manuscript is ready for publication.

Reviewer 1 ·

Basic reporting

No futher comment

Experimental design

No futher comment

Validity of the findings

No futher comment

Additional comments

No futher comment

Reviewer 2 ·

Basic reporting

No comment

Experimental design

No Comment

Validity of the findings

No Comment

Additional comments

No Comment